# Cerebrospinal Fluid Chloride Is Associated with Disease Activity of Relapsing–Remitting Multiple Sclerosis: A Retrospective Cohort Study

**DOI:** 10.3390/brainsci13060924

**Published:** 2023-06-07

**Authors:** Xingwei Fang, Yaxin Lu, Yongmei Fu, Zifeng Liu, Allan G. Kermode, Wei Qiu, Li Ling, Chunxin Liu

**Affiliations:** 1Faculty of Medical Statistics, School of Public Health, Sun Yat-sen University, Guangzhou 510080, China; fangxw8@mail2.sysu.edu.cn; 2Clinical Data Center, The Third Affiliated Hospital of Sun Yat-sen University, Guangzhou 510630, China; luyx68@mail.sysu.edu.cn (Y.L.); liuzf@mail.sysu.edu.cn (Z.L.); 3Emergency Department, The Third Affiliated Hospital of Sun Yat-sen University, Guangzhou 510630, China; fuyongmei1973@163.com; 4Perron Institute, University of Western Australia, Nedlands, WA 6009, Australia; 5Neurology Department, The Third Affiliated Hospital of Sun Yat-sen University, Guangzhou 510630, China; qiuwei@mail.sysu.edu.cn; 6Clinical Research Design Division, Clinical Research Centre, Sun Yat-sen Memorial Hospital, Sun Yat-sen University, Guangzhou 510120, China

**Keywords:** multiple sclerosis, cerebrospinal fluid chloride, relapse

## Abstract

Background: Blood-brain barrier dysfunction in active multiple sclerosis (MS) lesions leads to pathological changes in the cerebrospinal fluid (CSF). This study aimed to investigate the possible association between routine CSF findings, especially CSF chloride, at the time of the first lumbar puncture and the relapse risk and disability progression of relapsing–remitting MS (RRMS). Methods: This retrospective study included 77 patients with RRMS at the MS Center of our institution from January 2012 to December 2020. The Anderson and Gill (AG) model and Spearman correlation analysis were used to explore predictors of relapse and disability during follow-up. Results: In the multivariate AG model, patients with elevated CSF chloride level (hazard ratio [HR], 1.1; 95% confidence interval [CI]: 1.06–1.22; *p* = 0.001) had a high risk of MS relapse. Using median values of CSF chloride (123.2 mmol/L) as a cut-off, patients with CSF chloride level ≥ 123.2 mmol/L had a 120% increased relapse risk compared with those with CSF chloride level < 123.2 mmol/L (HR = 2.20; 95% CI: 1.19–4.05; *p* = 0.012). Conclusions: Elevated CSF chloride levels might be a biologically unfavorable predictive factor for disease relapse in RRMS.

## 1. Introduction

Multiple sclerosis (MS) is a chronic, inflammatory, demyelinating, and neurodegenerative disease of the central nervous system (CNS). It is the primary cause of non-traumatic neurological disability among young adults, with a higher prevalence observed in women. The prevalence of MS varies globally, with the highest prevalence reported in northern regions and lower prevalence observed in equatorial countries. Recent data suggest that the prevalence of MS has been on the rise over the last few decades [1].

MS has three distinct clinical courses: relapsing–remitting MS (RRMS), secondary progressive MS (SPMS), and primary progressive MS (PPMS). In addition, clinically isolated syndrome (CIS) often progresses to MS, especially when symptoms are accompanied by CNS lesions. RRMS is the most common phenotype [2] and is characterized by unpredictable clinical relapses and remissions and gradual disability progression [3]. Early RRMS is usually marked by acute episodes of neurological deficits known as relapses, which rely on the location of the CNS region affected by the acute inflammatory demyelinating lesions and the extent of the inflammatory process.

Relapses are associated with a considerable increase in economic costs as well as a deterioration in health-related quality of life and functional abilities, which are some of the biggest concerns associated with the disease [4]. RRMS causes the accumulation of irreversible disability over time, further increasing the burden on patients and society [5].

Currently, RRMS remains incurable due to the unclear pathogenesis. However, with the development of increasingly effective therapies for RRMS, disease-modifying therapies could modify the course of disease through suppression or modulation of immune function [6,7]. They exert anti-inflammatory activity primarily in the relapsing phase of MS, reduce the rate of relapses, reduce accumulation of MRI lesions, and stabilize, delay, and in some cases modestly improve, disability. Nonetheless, the clinical course of RRMS varies greatly across patients and is unpredictable at the initial disease stages, making the treatment challenging for clinicians. Therefore, exploring effective biomarkers to predict disease activity over time is a prerequisite to creating a therapeutic algorithm for patients presenting with a highly active course in the early stages of the disease.

The diagnosis of MS typically involves a comprehensive evaluation of the patient’s medical history and a neurological examination, which aims to determine if there is dissemination in time and space of certain clinical symptoms while excluding other conditions with similar symptoms. Supportive diagnostic evidence may be obtained through paraclinical tests including Magnetic Resonance Imaging (MRI), evoked potential studies, and analysis of cerebrospinal fluid (CSF). These tests can provide supportive diagnostic information that helps clinicians confirm or rule out a diagnosis of MS [8]. Recently, the McDonald diagnostic criteria have been updated to allow the diagnosis of clinically definite MS (CDMS) after a single typical attack in patients who have MRI evidence of dissemination in space and the presence of oligoclonal bands (OCBs) in the CSF [3]. The new diagnostic guidelines underscore the significance of OCBs and highlight the growing importance of examining CSF.

Cerebrospinal fluid (CSF) analysis is a diagnostic or evaluative tool for many conditions affecting the CNS [9]; it has been proven that the presence of OCBs in patients with clinically isolated syndrome doubles the risk of a second clinical attack [10]. Theoretically, blood-brain barrier dysfunction in active MS lesions leads to pathological changes in the CSF, which could help predict disease relapse and disability progression [11]. Attempts have been made to correlate CSF parameters and clinical aspects of the disease in various studies, such as that of Lotan et al. [12], who reported that CSF pleocytosis at presentation is associated with a higher annual relapse rate and a more rapid progression of neurologic disability. Albanese et al. [13] found that CSF lactate is associated with MS disease progression. Diem et al. [14] found that increased CSF protein concentration have a negative impact on EDSS after relapse. CSF immunoglobulin free light chains, CSF neurofilament levels, and other CSF biomarkers have also been proposed as prognostic biomarkers [15,16].

CSF chloride is a measure of the concentration of chloride ions present in the cerebrospinal fluid that surrounds the brain and spinal cord. It is an important electrolyte in the CSF and plays a role in maintaining the proper balance of fluids and electrolytes in the central nervous system. Abnormal levels of CSF chloride may indicate underlying neurological conditions [17,18] such as meningitis, encephalitis, or hydrocephalus. However, very few studies have assessed the association between the levels of CSF chloride and MS disease activity.

Chloride ions are the primary permeant anions in mammals, and homeostasis of their flux across cell membranes is important for a range of neurophysiological processes [19,20,21]. The concentration of chloride ions in neurons is very low; thus, small changes in the concentration greatly impact the transmembrane gradient [22]. Several studies have highlighted that abnormal chloride homeostasis is associated with neuronal trauma or brain disorders such as hypoxic–ischemic encephalopathy, brain edema, and post-traumatic seizures [23,24,25]. The role of chloride ions in the pathogenesis of MS has not been fully elucidated; however, emerging human and animal models directly link intra-myelinic edema and ion homeostasis [26].

Therefore, this study aimed to investigate the possible association between routine CSF findings (at the time of the first lumbar puncture [LP] during the initial presentation), especially CSF chloride, and the relapse risk and disability progression of MS.

## 2. Materials and Methods

### 2.1. Study Design and Participants

This retrospective study, which followed the STROBE guidelines [27], was approved by the Ethics Committee of the Third Affiliated Hospital of Sun Yat-sen University (grant number: [2020]02-423), and all participants provided written informed consent.

We retrospectively reviewed the clinical data of 335 patients with MS who attended the Neurology Department of the Third Affiliated Hospital of Sun Yat-sen University from January 2012 to December 2020. We included patients of all ages who fulfilled all the following criteria: diagnosed with RRMS according to the 2017 modified McDonald criteria [3]; without diagnosis of other systemic diseases; with at least one MS clinical episode; and had undergone LP during the initial diagnostic process. Patients with a different type of MS, those without LP, and those with incomplete information were excluded. As of the end of the study, no patients had died or moved. Figure 1 shows a flowchart of the inclusion and exclusion of data.

We collected the demographic and clinical data of eligible patients, including age at disease onset, sex, disease course phenotype, maintenance therapy, subsequent relapse (start and stop date, outcome), expanded disability status scale (EDSS) score, and laboratory examination data. Baseline characteristics were collected when patients were admitted for treatment due to a first attack or relapse during the study period and all relapses were recorded and archived by patient medical records during the follow-up period. For an individual subject, regular follow-up is set at 3 months, 6 months, and 1 year after the first day of MS diagnosis. If symptoms are stable, maintain annual follow-up. At least one senior neurologist calculated the EDSS score throughout the disease duration period.

CSF was evaluated for the number of white blood cells, protein (g/L), chlorine and glucose levels (mmol/L), and quotient of CSF and serum glucose, chlorine, and protein levels. For analysis, maintenance therapies were classified into two categories: disease-modifying therapies (DMTs) such as interferon beta, fingolimod, mitoxantrone, rituximab, and dimethyl fumarate, and non-DMTs, such as azathioprine, mycophenolate mofetil, methotrexate, tacrolimus, cyclophosphamide, corticosteroids, and intravenous immunoglobulin administration. The duration of MS was calculated from the time of the patient-reported first clinical manifestation of the disease.

### 2.2. Outcome Definition

Attack/relapse was defined as a monophasic clinical episode with patient-reported symptoms, increase in EDSS, and objective findings reflecting a focal or multifocal inflammatory demyelinating event in the central nervous system, developing acutely or sub-acutely, with a duration of at least 24 h, with or without recovery, and in the absence of fever or infection [3]. ‘First attack’ was defined as the inaugural attack that marked the disease onset and ‘recurrent relapses’ as all attacks following the first attack.

Disease progression was evaluated by calculating multiple sclerosis severity score (MSSS), which is an algorithm that relates EDSS scores to the distribution of disability in patients with comparable disease duration [28].

### 2.3. Statistical Analysis

Baseline characteristics, including patient demographics, baseline EDSS score, annualized relapse rate, laboratory values at the time of the first LP, time between the first attack and LP, and maintenance therapies, are presented as counts and percentages for discrete variables and median (interquartile range, IQR) for continuous data. The correlation between the MSSS and CSF findings for the entire study population was calculated using the Spearman correlation coefficient. Survival curves were generated by estimating the cumulative hazard to reach the first relapse for RRMS patients stratified by the median of CSF findings using the Kaplan–Meier method expressing 95% confidence intervals (CIs) with the Survminer package [29].

To assess the risk of relapse over the entire disease course in patients, the Anderson and Gill model was used using the survival package [30,31,32]. Univariate and multivariate analyzes were performed, and multivariate models were adjusted for known MS prognostic markers, including demographic markers (sex and age at disease onset), and maintenance therapies [33,34,35].

All statistical analyzes were performed using R version 4.0.3 (The R Foundation for Statistical Computing, Vienna, Austria; http://www.r-project.org/ (accessed on 1 November 2020)), and *p*-values < 0.05 were considered significant.

## 3. Results

### 3.1. Demographics and CSF Results at the Time of the First LP

The cohort included 77 patients (50 women and 27 men; sex ratio, 1.85:1) who underwent LP at the first clinical episode, with a mean age of 28.8 ± 12.6 years. The mean follow-up period was 57.2 ± 19.7 months (range: 22.1–97.7 months).

At the time of the first LP in our cohort, the CSF chloride levels were in the range of 117.4–130.0 mmol/L. The range of the CSF leukocyte count was 0–28.0 cells/µL and correlated with CSF chloride level (*r_s_* = 0.23; *p* = 0.040). No significant difference was noted in CSF findings between male and female patients. The demographics and CSF results at the time of the LP are summarized in Table 1.

### 3.2. CSF Findings and MS Relapse Risk

In addition to the index events, 112 subsequent demyelinating relapses were registered, with a median number of relapses of 2 (IQR: 1, 3). Overall, 45 patients (58.4%) had at least one relapse, and Figure 2 shows the number of relapses along the disease course.

From univariate analysis, the risk of relapse decreased with the age of onset. Patients not treated with DMT had a greater risk of relapse over the entire disease course than those treated with DMT (Table 2). In addition, elevated CSF chloride level, CSF glucose level, serum protein level, and protein quotient were associated with a higher risk of relapse (Table 3).

In the multivariate analysis, the model was adjusted by including age at onset, sex, and maintenance treatment. Elevated CSF chloride level, CSF glucose level, CSF white blood cell (WBC) count, serum protein level, and protein quotient level were associated with a high risk of relapse. However, the remaining serum and quotient indicators had no significant effect on the risk of relapse (Table 3).

### 3.3. Cumulative Hazard Resulting in the First Relapse

To distinguish the potential relapse risk of patients with MS, we used the median values of CSF findings as cut-off values. Patients with CSF chloride level ≥ 123.2 mmol/L had a 120% increased risk of relapse compared with those with CSF chloride level < 123.2 mmol/L (HR = 2.20; 95% CI: 1.19–4.05; *p* = 0.012). Patients with CSF WBC count ≥ 2 cells/µL had an 84% increased risk of relapse compared with those with CSF WBC count < 2 cells/µL (HR = 1.84; 95% CI: 1.02–3.32; *p* = 0.043). Patients with CSF glucose level ≥ 3.3 mmol/L had a 141% increased risk of relapse compared with those with CSF glucose level < 3.3 mmol/L (HR = 2.41; 95% CI: 1.30–4.49; *p* = 0.006). The cumulative hazard of the relapse events is shown in Figure 3.

### 3.4. CSF Findings and Disease Course

Next, we calculated the MSSS for each patient at the time of sampling and the first relapse, and correlated the data with the initial CSF findings. Significant correlation was only determined between CSF WBC and the MSSS in the patients with MS (Table 4). When stratifying by maintenance therapies, we also found a positive correlation between CSF WBC and MSSS in the not-DMT group (Appendix A).

### 3.5. CSF Findings and Presence of CSF OCBs at Presentation

The relationship between CSF findings and the presence of CSF OCBs at presentation was also examined. CSF OCBs were present in 49.4% of the 77 patients included in the study. There was no significant difference between the CSF findings of patients with positive OCBs and those without positive OCBs (Appendix A). Elevated CSF chloride level, CSF WBC count, and CSF glucose level were associated with a high risk of relapse in the multivariate stratifying patients by the presence or absence of OCB (Appendix A).

## 4. Discussion

In this monocentric and retrospective study from 2012 to 2020, we analyzed the data of patients with RRMS who underwent their first LP and confirmed that an increased CSF chloride concentration has a negative impact on the risk of relapse. This finding may have therapeutic implications and indicates the need for a more aggressive therapeutic approach.

A previous study substantiated the concept that chloride channels are essential for ion and water homeostasis in the brain. Astrocytes are pivotal to maintaining ion and water homeostasis in the brain and play a significant role in potential action generation and impulse conduction [36]. Loss of function of the chloride channel is linked to brain white matter edema, which radiological correlates and functional models have clearly demonstrated [26].

Interestingly, using the Anderson and Gill model to assess the risk of recurrent events, we found that patients with elevated CSF chloride levels were at a high risk of relapse. Furthermore, we observed that the relapse risk of patients with CSF chloride level ≥ 123.2 mmol/L was more than double that of patients with CSF chloride level < 123.2 mmol/L for the first relapse, when using median CSF chloride concentrations as the cut-off point. This finding provides useful information for individual patients tailor the specific agent to address a more active course. The ability of elevated CSF chlorides at the time of the first LP to predict disease relapse could be correlated with the breakage of ions and water homeostasis in the brain, which might impact action potential generation and impulse conduction. Elevated CSF chlorides might also represent an innate response to infection, probably viral, triggering an initial relapse; whether they represent a marker or a factor of pathogenic significance is still unknown.

Two other covariates, age at onset and maintenance treatment approach, were associated with the risk of relapse, which was in accordance with previous studies: increasing age was associated with fewer relapses [35,37]. The decline in clinical and subclinical disease activity with aging is a clinical phenomenon strongly associated with the progressive loss of the innate and adaptive immune system’s efficiency with aging, which may be triggered by an adaptive or a dysregulated response [38]. Currently, several DMTs are available for the treatment of RRMS and are considered to reduce the immune-mediated inflammatory process of the central nervous system, which translates into demonstrable improvements in clinical and radiological outcomes [39]. The present study found that DMT is superior to no-DMT for relapse control, which indicates that a suitable therapeutic regime is more favorable for relapse control.

In addition, we found that an increased leukocyte count in CSF and increased protein quotient were also related to a higher risk of relapse, which is consistent with the recent reports [12,14]. We also observed a positive correlation between leukocyte count in CSF and MSSS, especially in patients who did not receive DMT. These results suggest that an increased leukocyte count in CSF may reflect a more active disease in patients, highlighting the importance of early DMT. Perturbed glucose metabolism is implicated in neurodegenerative disorders like Alzheimer’s, Parkinson’s, and Huntington’s. However, little is known about its role in MS pathology. We found that an increase in glucose level in the CSF was also related to a higher risk of relapse, which may be related to mitochondrial aberrations and impaired glucose metabolism in MS [40].

Remarkable advancements have been made in the treatment of MS, thanks to an improved understanding of its pathogenesis and course. Immunosuppressive therapies have proven effective for controlling relapses and delaying disability progression in patients with RRMS [41]. In the past, high-dose IFN-b-1a and glatiramer acetate were typically preferred as first-line options; however, recent studies recommend using highly effective disease-modifying therapies as the initial approach for most patients with active RRMS [6]. This approach supersedes the traditional “treat to target” method which involves using a moderately effective treatment initially and then advancing to a more potent agent when breakthrough disease occurs. Observational studies suggest that early use of high-efficacy therapy can lead to better long-term outcomes. However, due to significant disease heterogeneity, clinical and biological markers are necessary to develop personalized treatment plans based on each patient’s specific risk of relapse and disability progression. 

As diagnostic criteria for MS continue to improve, CSF testing has become increasingly important [3]. In addition to assisting in the differential diagnosis of neurological disorders, CSF tests are significant sources of supportive diagnostic evidence. OCBs in the CSF are present in up to 90% of MS patients [42], while various studies have identified other CSF biomarkers associated with MS relapse and disability progression [12,14,43]. This study found that CSF chloride, glucose, and white blood cell count at the first onset were associated with a higher risk of relapse, which may predict a more active disease course and eventually indicate the need for a more active therapeutic approach.

This study has some limitations. The primary limitations of this study are its retrospective nature and the associated possibility of recall bias. Additionally, we could not further confirm the correlation with the clinical results due to the lack of magnetic resonance imaging and paraclinical data. Thirdly, the sample size of our study was relatively small, and we only included Asian populations. Lastly, due to the limited sample size, further categorization of DMT into first-line and second-line drugs was not possible. Further prospective studies with larger sample sizes are required to validate our findings.

## 5. Conclusions

Patients with RRMS with elevated CSF chloride level at the time of the first LP (especially those with CSF chloride level ≥123.2 mmol/L) were likely to show a high risk of relapse, indicating the need for a more active immunosuppressive treatment.

## Figures and Tables

**Figure 1 brainsci-13-00924-f001:**
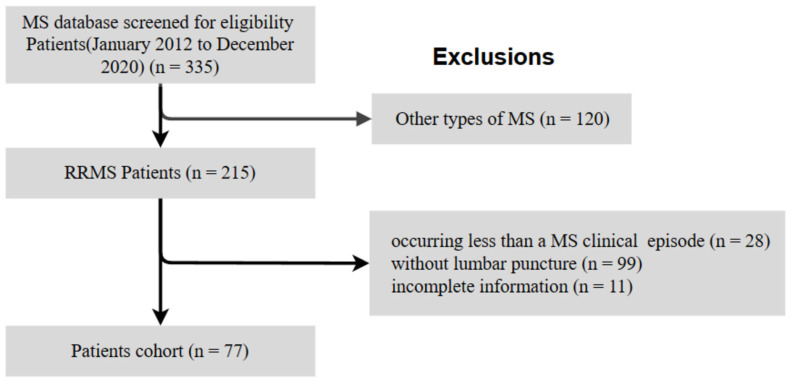
Flowchart of the study cohort. MS, multiple sclerosis; RRMS, relapsing–remitting multiple sclerosis.

**Figure 2 brainsci-13-00924-f002:**
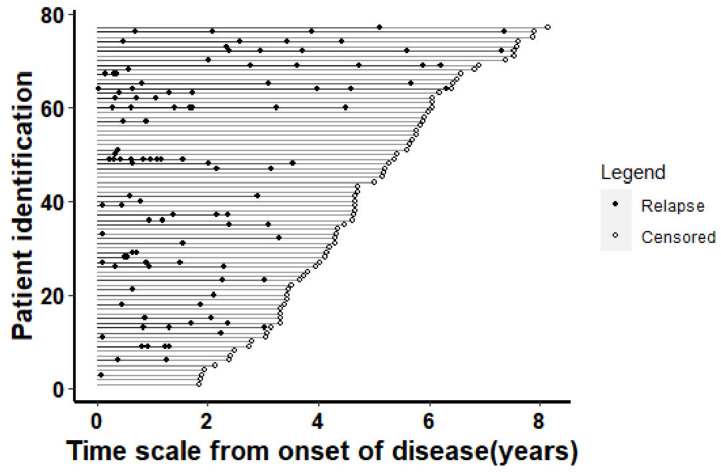
Relapse distribution along the whole disease cohort.

**Figure 3 brainsci-13-00924-f003:**
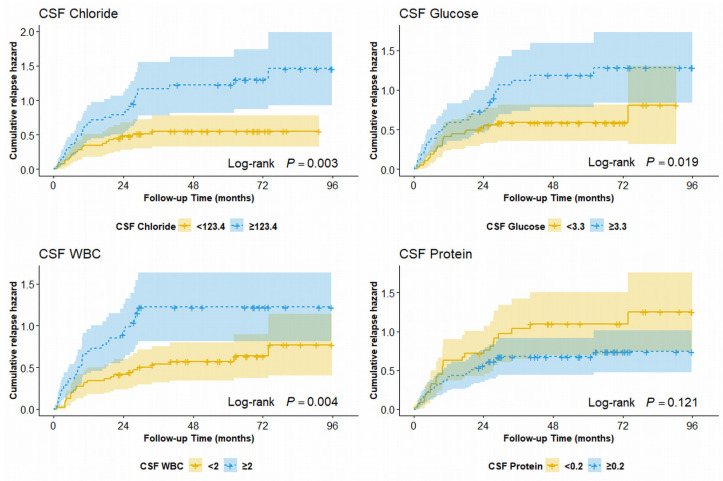
Cumulative hazard of first relapse for RRMS patients stratified by median of CSF findings. RRMS, relapsing–remitting multiple sclerosis; CSF, cerebrospinal fluid.

**Table 1 brainsci-13-00924-t001:** Demographic and clinical features of relapsing–remitting multiple sclerosis patients.

Variable	RRMS Patients (*n* = 77)
Age at disease onset (years)	27.0 [22.0, 34.0]
Sex (%)	
Male	27.0 (35.1)
Female	50.0 (64.9)
Baseline EDSS score	2.5 [1.5, 3.5]
Maintenance therapies ^a^ (%)	
Disease-modifying therapies	28.0 (36.4)
No disease-modifying therapy	49.0 (63.6)
Annualized relapse rate	0.5 [0.2, 0.7]
Follow-up time (months)	55.9 [41.2, 71.0]
Time between first attack and LP (months)	22.3 [3.4, 57.2]
Time between first attack and therapy	37.0 [7.0, 382.0]
Oligoclonal band (%)	
Oligoclonal band +	38.0 (49.4)
Oligoclonal band-	39.0 (50.6)
CSF data	
CSF chloride (mmol/L)	123.2 [121.5, 124.5]
CSF glucose (mmol/L)	3.3 [3.1, 3.8]
Number of CSF WBCs (cells/µL)	2.0 [0.0, 6.0]
CSF protein (mmol/L)	0.2 [0.2, 0.3]
Serum data	
Glucose (mmol/L)	4.7 [4.3, 5.7]
Chloride (mmol/L)	104.5 [102.4, 105.5]
WBC count, ×10^3^ (μL)	7.1 [5.8, 9.4]
Protein (g/L)	64.4 [61.4, 67.5]
Ratio	
CSF/serum chloride	0.8 [0.8, 0.9]
CSF/serum glucose	1.5 [1.3, 1.7]
CSF/serum protein	287.4 [215.0, 379.0]

Values are presented as n (%) or median [interquartile range]. ^a^ All patients were treated after lumbar puncture. RRMS, relapsing–remitting multiple sclerosis; EDSS, expanded disability status scale; LP, lumbar puncture; CSF, cerebrospinal fluid; WBC, white blood cells.

**Table 2 brainsci-13-00924-t002:** Univariate analysis for adjusted variables.

Variable	HR (95% CI)	*p*-Value
Age at disease onset	0.98 (0.95–0.99)	0.041
Sex		
Male	Reference	-
Female	1.11 (0.62–1.99)	0.719
Maintenance therapies		
Disease-modifying therapy	Reference	-
No disease-modifying therapy	2.35 (1.44–3.83)	0.001

HR, hazard ratio; CI, confidence interval.

**Table 3 brainsci-13-00924-t003:** Univariate and multivariate Cox regression analyzes for recurrent events.

Variable	Univariate		Multivariate ^a^	
	HR (95% CI)	*p*-Value	HR (95% CI)	*p*-Value
CSF chloride	1.16 (1.06–1.27)	0.001	1.14 (1.06–1.22)	0.001
CSF glucose	1.45 (1.15–1.82)	0.002	1.46 (1.23–1.74)	<0.001
CSF protein	0.29 (0.02–5.43)	0.407	0.97 (0.05–17.16)	0.983
CSF WBC	1.04 (0.99–1.10)	0.108	1.06 (1.02–1.11)	0.004
Serum chloride	1.02 (0.89–1.16)	0.784	1.01 (0.91–1.11)	0.895
Serum glucose	1.05 (0.88–1.25)	0.601	1.02 (0.89–1.18)	0.780
Serum protein	1.05 (1.01–1.09)	0.021	1.05 (1.01–1.08)	0.006
Serum WBC	0.99 (0.90–1.08)	0.776	0.98 (0.92–1.05)	0.647
CSF/serum chloride	0.01 (0.01–271.02)	0.249	0.01 (0.01–9.71)	0.128
CSF/serum glucose	0.70 (0.34–1.44)	0.330	0.60 (0.34–1.07)	0.081
CSF/serum protein	1.00 (1.00–1.00)	0.007	1.00 (1.00–1.00)	0.047

^a^ The multivariate model was adjusted for sex and age at disease onset, and maintenance therapies. HR, hazard ratio; CI, confidence interval; CSF, cerebrospinal fluid; WBC, white blood cells.

**Table 4 brainsci-13-00924-t004:** Correlation between CSF findings and the Multiple Sclerosis Severity Score at the time of the first lumbar puncture and first relapse.

Variable	First Lumbar Puncture		First Relapse	
	Correlation Coefficient	*p*-Value	Correlation Coefficient	*p*-Value
CSF chloride	0.12	0.281	0.02	0.872
CSF glucose	0.13	0.243	0.17	0.137
CSF WBC	0.26	0.021	0.16	0.153
CSF protein	0.07	0.542	0.02	0.857
Serum chloride	−0.10	0.364	0.02	0.872
Serum glucose	0.07	0.537	0.17	0.137
Serum protein	0.02	0.885	0.16	0.153
Serum WBC	0.03	0.782	0.02	0.857
CSF/serum chloride	−0.07	0.521	−0.22	0.052
CSF/serum glucose	−0.06	0.631	−0.03	0.787
CSF/serum protein	0.12	0.281	0.02	0.872

WBC, white blood cell; CSF, cerebrospinal fluid.

## Data Availability

The datasets generated during and/or analyzed during the current study are available from the corresponding author on reasonable request.

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
