# Peer review of "Cerebrospinal Fluid Chloride Is Associated with Disease Activity of Relapsing–Remitting Multiple Sclerosis: A Retrospective Cohort Study"

_brainsci, 2023, doi:10.3390/brainsci13060924_

Round 1
Reviewer 1 Report
Comments and Suggestions for Authors
Introduction:
Please provide more background on MS
Please clearly state the hypothesis
Methods:
Does your study been reviewed and approved by an ethic committee?
On table 1, you present only female but in results 3.1 you are mentioning males.
It's misleading, the statistics were made on the total population or just on males or females? If it's on the entire population could you please have a comparison males and females ?
Results: The last item of the flow chart states patient cohort 115 but on table 1 we can see patient n= 77. can you clarify please.
Thanks
Reviewer 2 Report
Comments and Suggestions for Authors
I read the manuscript titled "Cerebrospinal fluid chloride is associated with disease activity of relapsing-remitting multiple sclerosis: a retrospective cohort study" with interest because of the topic addressed in the search for reliable predictors of relapsing-remitting multiple sclerosis. I think this is an extremely interesting issue and the results are needed.
Below are the comments that occurred to me during the analysis of the manuscript:
1. Section "Abstract": please insert the explanation of the abbreviation MS (line 16), as it is only explained in the "Introduction" section.
2. Sub-section 2.1: are all inclusion and exclusion criteria listed? Were age and the presence of other diseases not included among these criteria? Did the use of patients' data not require their formal consent?
3. The first sentence of line 108 ("Figure 1 shows a flowchart of the inclusion and exclusion of data.") as well as Figure 1 should be moved to the "Materials and Methods" section. Additionally, regarding Figure 1: a) please insert an explanation of the abbreviations ("MS" and "RRMS") and b) the size of the study group is inconsistent with the information provided in the "Abstract" (n=77, line 20) and in subsection 3.1 regarding the results ("The cohort included 77 patients ...", lines 107-108). The figure states "Patient cohort (n=115)". I will ask you to clarify and standardize in the manuscript the final number of subjects whose data were used in the study.
4. Table 1: I will ask you to modify the heading to include the word "patient" or "subjects" (in other words, please indicate that this table is about the characteristics of the subjects), as well as please for some "Variables" to complete the unit in which they are presented (e.g., this information is missing for "Female" - is it "n (%)"?).
5. The insertion of the reference to the literature item in line 190 should be improved (currently there is "[22,22-24]". Isn't "[22-24]" enough?
6. Another issue concerns the entire section presenting the results of the study and the discussion of the results. One wonders why the authors focused on chloride levels in CSF? In addition, the inference is only about this result. I understand that their specific role and metabolism in the central nervous system is known, but the authors should justify why only this parameter remained their focus. Unfortunately, such an indication of one relationship suggests that the others were irrelevant, which is probably not true? In addition, the entire passage of text from the "Discussion" section a regarding the role of chloride ions (lines 184-200) should be moved to the "Introduction" section with a concomitant justification by the authors regarding the interest mainly in chlorides. Moreover, in the "Discussion" section in lines 192-194, the authors wrote that "Our study aimed to examine the relationship between disease activity, measured by calculated relapse risk, and accumulating neurological disability and CSF chloride at the time of the first LP. ", while the objective stated in the "Abstract" and "Introduction" sections reads differently ("This study aimed to investigate the possible association between routine CSF findings at the time of the first lumbar puncture and the relapse risk and disability progression of relapsing-remitting-multiple sclerosis (RRMS)." - lines 16-18 of the "Abstract" section. "Therefore, this study aimed to investigate the possible association between routine CSF findings (at the time of the first lumbar puncture [LP] during the initial presentation) and the relapse risk and disability progression of MS." - lines 51-53 of the "Introduction" section.
7. In the "Discussion" section, in the section on limitations of the study, the authors included the following statement: "Thirdly, the sample size of our study was relatively small, and we only included Asian populations [36], ..." (lines236-237). Shouldn't the authors then have included ethnic group in the study's inclusion criteria? And why does this part of the sentence have a reference to literature items?
8. The "References" section includes more than 60% of "old" literature items, i.e., from the last five years ago - could this section be updated, at least in part?
Reviewer 3 Report
Comments and Suggestions for Authors
This study is an interesting study that aimed to explore the potential correlation between routine cerebrospinal fluid obtained during the initial lumbar puncture and the risk of relapse as well as the progression of disability in individuals with relapsing-remitting multiple sclerosis. The study is both original and relevant, making an important contribution to the field. However, I understand that the manuscript needs to be revised and some suggestions and questions are presented here for good application of findings.
1. Title: Adequate
2. Abstract: The abstract section is clear and it explains the main findings of the study in accordance with the journal's guidelines (up to 200 words)
3. Keywords: Although not mandatory, it is recommended that you use MeSH (Medical Subject Headings) terms.
4. Introduction: The introduction of the study should be more detailed. The theme is poorly introduced, lacking appropriate information regarding the disease, the diagnostic procedures, and the overarching topic/question addressed in this study. Only seven references were used in this section, which could be improved. Including more current articles could make the introduction more engaging for the reader.
5. Methods: Please explain the rationale for delimiting the study period to only 8 years (2012 to 2020). Justify the decision to conduct a retrospective study instead of a prospective study. No patient died or moved during the 8-year follow-up period? The flowchart does not demonstrate this. Include information regarding disease duration of time since diagnosis. Cognition was not assessed in this study? Provide an explanation for performing correlation analyses between cerebrospinal fluid (CSF) and the MS Severity Score.
6. Results: The results are thoroughly and comprehensively presented. Congratulations on the meticulous detailing of the findings!
7. Discussion: Not all patients undergo CSF examination. How could this finding be useful for this population? Please provide a more detailed discussion from the perspective of active immunosuppressive treatment.
8. Conclusion: Adequate
9. Acknowledgements: Correct or exclude this section.
10. References: Adequate
Comments on the Quality of English LanguageMinor editing of English language required
Round 2
Reviewer 2 Report
Comments and Suggestions for Authors
I thank the authors for responding to my comments and suggestions and taking them into account in the development of the revised version of the manuscript.